# Thermal Performance of Single-Story Air-Welled Terraced House in Malaysia: A Field Measurement Approach

**Pau Chung Leng** [1,*] **, Gabriel Hoh Teck Ling** [1] **, Mohd Hamdan Ahmad** [1] **, Dilshan Remaz Ossen** [2] **, Eeydzah Aminudin** [3] **, Weng Howe Chan** [4] **and Dg Normaswanna Tawasil** [3]

[1] Faculty of Built Environment & Surveying, Universiti Teknologi Malaysia, Johor 81300, Malaysia; gabriel.ling@utm.my (G.H.T.L.); b-hamdan@utm.my (M.H.A.)

[2] Department of Architecture Engineering, Kingdom University, Riffa 40434, Bahrain; d.ossen@ku.edu.bh

[3] School of Civil Engineering, Faculty of Engineering, University Teknologi Malaysia, Johor 81300, Malaysia; eeydzah@utm.my (E.A.); wannatawasil@gmail.com (D.N.T.)

[4] School of Computing, Faculty of Engineering, Universiti Teknologi Malaysia, Johor 81300, Malaysia; cwenghowe@utm.my

\* Correspondence: pcleng2@utm.my; Tel.: +60-7-553-7389

**Abstract:** The provision requirement of 10% openings of the total floor area stated in the Uniform Building By-Law 1984 Malaysia is essential for natural lighting and ventilation purposes. However, focusing on natural ventilation, the effectiveness of thermal performance in landed residential buildings has never been empirically measured and proven, as most of the research emphasized simulation modeling lacking sufficient empirical validation. Therefore, this paper drawing on field measurement investigates natural ventilation performance in terraced housing with an air-well system. The key concern as to what extent the current air-well system serving as a ventilator is effective to provide better thermal performance is to be addressed. By adopting an existing single-story air-welled terrace house, indoor environmental conditions and thermal performance were monitored and measured using HOBO U12 air temperature and humidity, the HOBO U12 anemometer, and the Delta Ohm HD32.3 Wet Bulb Globe Temperature meter for a six-month duration. The results show that the air temperature of the air well ranged from 27.48 °C to 30.92 °C, with a mean relative humidity of 72.67% to 79.25%. The mean air temperature for a test room (single-sided ventilation room) ranged from 28.04 °C to 30.92 °C, with a relative humidity of 70.16% to 76.00%. These empirical findings are of importance, offering novel policy insights and suggestions. Since the minimum provision of 10% openings has been revealed to be less effective to provide desirable thermal performance and comfort, mandatory compliance with and the necessity of the bylaw requirement should be revisited.

**Keywords:** air shaft; solar chimney; air well; field measurement; natural ventilation; tropical climate; terrace house; passive cooling design

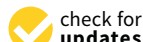

## 1. Introduction

Natural ventilation occurs when pressure differences generated by wind or buoyancy forces undertake at single or multiple openings in the building envelope. It is an important and significant sustainable building design strategy for human being as one of the basic living environment criteria [1–5]. Following the Malaysia Uniform Building By-Law 1984, the requirement of provisioning a minimum of 10% of the total floor area of residential and business spaces has to be fulfilled in order to secure approval and commence a construction process on the ground. A similar requirement has been stated in the Building and Construction Act of Singapore as well as in other countries such as Australia and some other Commonwealth countries. Hence, it is plausible that natural ventilation is important and necessary to be included as a habitable building design strategy under circumstances when a mechanical ventilation system is unavailable.

In Malaysia, terraced houses have a limited amount of exposed building envelope due to the constraint of design layout with adjacent party walls. Such constraints have limited the natural ventilation strategy to be applied on the external fenestration. Hence, mechanical ventilation systems have become the choice of occupants in a terraced house as a substitution to manage air quality and indoor ventilation and subsequently solve the thermal comfort problem. However, mechanical ventilation systems require high electricity to run. For instance, in some cities, air-conditioning requirements take the full capacity of electricity grids [3,6]. According to Toe and Kubota [7], final energy use in residential and commercial sectors in Malaysia increased more than sevenfold between 1980 and 2007, at a higher rate than the total amount of energy-demand growth rate, based on the Ministry of Energy, Green Technology, and Water National Energy Balance 2007 report. In residential buildings, a large portion of electricity consumption was attributed to air-conditioning systems, as the ownership of air-conditioners expanded from approximately 12,000 to 764,000 households from the 1970s to the 2000s [7,8].

In recent years, the natural ventilation topic has attracted growing interest due to its potential benefits over mechanical ventilation systems in terms of economic, energy consumption, and environmental advantages [6]. There were also some trials to enhance the performance of natural ventilation by using renewable energy, whereas some authors have categorized it as hybrid ventilation [5]. This has proven that natural ventilation is widely researched for the purpose of achieving a reduction in the use of air-conditioning systems. An application of natural ventilation to provide thermal comfort is a sustainable approach because the method applied uses renewable energy resources, such as solar energy, and hence minimizes energy consumption in buildings [9].

The thermal performance of an indoor environment under a hot climatic condition is highly affected by various passive design techniques, for instance, space dimensions, facade colors, fenestration ratio, and glazing type, and vertical and horizontal shading devices [10]. The principle of an air-well effect is based on the combination of both solar-assisted stack ventilation as well as wind-driven ventilation. That is, solar heating causes hot air to rise, and due to the light density of the hot air, it escapes from shaft outlets. Meanwhile, the cooler air withdraws into the indoor environment from fenestrations via the pull effect complemented by the push effect from the outdoor environment [2,5,11]. In the past decades, much research was conducted on ventilation shaft configurations and strategies to improve a ventilating system: for instance, the improvement of solar chimney performance by using different types of glazing; increasing the air gap, width, depth, and height of the solar chimney; integrating the Trombe wall with a roof solar collector; and changing the inclination angle of the solar chimney [12–16].

Most of the above research regarding ventilation strategies was mainly focused on simulation methods, although some studies adopt and justify their simulation methodology, which is sufficiently deemed accurate and valid, and some have been conducted specifically on ventilation shafts focusing on the courtyard, solar chimney, and Trombe wall of domestic residential buildings [6,17–26]. However, due to data collection constraints in obtaining real data, only a few were validated with field measurement results. In other words, there is still a lack of empirical field research conducted, specifically investigating the effectiveness and impact of an air well (as a ventilator), especially in a tropical climate country such as Malaysia. More precisely, albeit the importance of natural ventilation is emphasized as it is also part of the legal requirement of the bylaw to provide the minimum percentage of opening (10% of total room floor area), the workability **or** practicality and effectiveness in terms of thermal performance of the minimum amount of the opening imposed (i.e., an air well) is still unknown because so far no single empirical research has been carried out in this regard. In Malaysia, terraced houses dominate the overall property record. According to the Summary of Property Market Report 2019, under the section of Overall Performance of Malaysia's Property Stock, Planned Supply and Incoming Supply, residential property recorded the highest figure of 5,727,814 units, followed by shops recording 526,079 units and service apartments with 253,056 units. Of the more than five million units of residential

types that were built, terraced houses comprised 40.9%, or 85,669 units, compared to other residential types, such as high-rise, vacant plot, semi-detached, or detached houses [27], Since terraced houses are dominant in Malaysia, primarily using an air-well system, it is therefore vital to understand the current ventilation and thermal performance of the existing room via field measurement. More importantly, the key concern as to what extent the current air-well system serving as a ventilator is effective to provide high thermal performance in a single-story terraced house is to be addressed.

This paper is novel as it adds value in several ways. Aside from providing theoretical insights on how the configurations (e.g., position, size, height, and shape) of an air well affect its ventilation performance, it primarily contributes methodologically via various data collection (i.e., step-by-step setting up of equipment) and analysis techniques, as well as empirically (a detailed case study via a field measurement analysis on the thermal performance of an air-well system), which hence provides pragmatic policy implications not only to the Malaysian housing architectural and construction fields but also to other parts of the world. This means that the significance of the thermal performance of the air shaft terraced house could be studied more extensively in the future based on the current field measurement results. In addition, this study is in line with the Sustainable Development Goals (SDG), since governments and all the agencies from profit or non-profit organizations are committed to achieving the goals in order to reduce urban energy consumption to 80% of global energy. Since buildings account for 40% of total energy consumption, designs of an energy-efficient building in passive designs (natural ventilation) could make a significant contribution to meeting SDG 11 and SDG 13 on climate action [28].

The remainder of the paper is structured as follows: Section 2 continues with a literature review that focuses on the effectiveness of the ventilation shaft and an empirical study of its thermal performance. Next, Section 3 discusses the methodology highlighting empirical field measurement techniques in terms of data collection and analysis, which were executed on the selected case study of a single-story terraced house. Section 4 presents empirical results and discussions on the thermal performance of an air well within the terraced house's test room. Lastly, Section 5 concludes by summarizing key findings (takeaways) and suggesting future research based on several limitations. In this paper, the terminology of "air shaft," "air well," and "solar chimney" are used interchangeably subject to different contexts, but essentially they entail the same thing.

## 2. Literature Review

### 2.1. Terraced Houses with Air-Well Systems

Prior to discussing how an air-well system vitally works as a ventilator for improved thermal performance in a residential building, a description of a terraced house is provided. In Malaysia, a typical single terraced house unit consists of a built-up area of 60 to 65 square meters with 6.5 m of front width and 11 m in length. A terraced house is usually sandwiched by two sides of party walls with minimum front and rear widths exposed to the external environment. The Uniform Building By-Law 1984 (UBBL 1984) states that a terraced house could be referred to as any residential building that is designed as a single dwelling unit, and forms part of a terrace of not less than three such residential buildings [29]. Terraced houses usually have a narrow frontage, and the party walls are shared with adjacent houses [30]. Internal partition walls within the narrow terrace housing layout define living spaces, such as living room, bedrooms, utility, bathroom, and kitchen. Thus, an air well could be used in one those spaces, and it is a typical feature in a traditional shophouse in Malaysia. It provides natural daylight and ventilation to the internal space. It is a vertical shaft or opening penetrated from roof to floor, connecting the internal space to the open sky. However, the size of an air-well opening is huge, which would lead to security issues. Thus, the feature has been replaced with an atrium or elevated clerestory in modern terraced houses. In order to improve thermal performance in terraced houses, air wells have been modified to solar chimneys, following Malaysian building regulations and thus keeping the house secured. Most of the time, the air well in a modern terraced house is located in

the intermediate room, utility room, or bathroom, as shown in Figure 1 [31]. It is the most versatile feature of traditional row houses, where the number of air wells may range from one to three or four based on the length of the house [32].

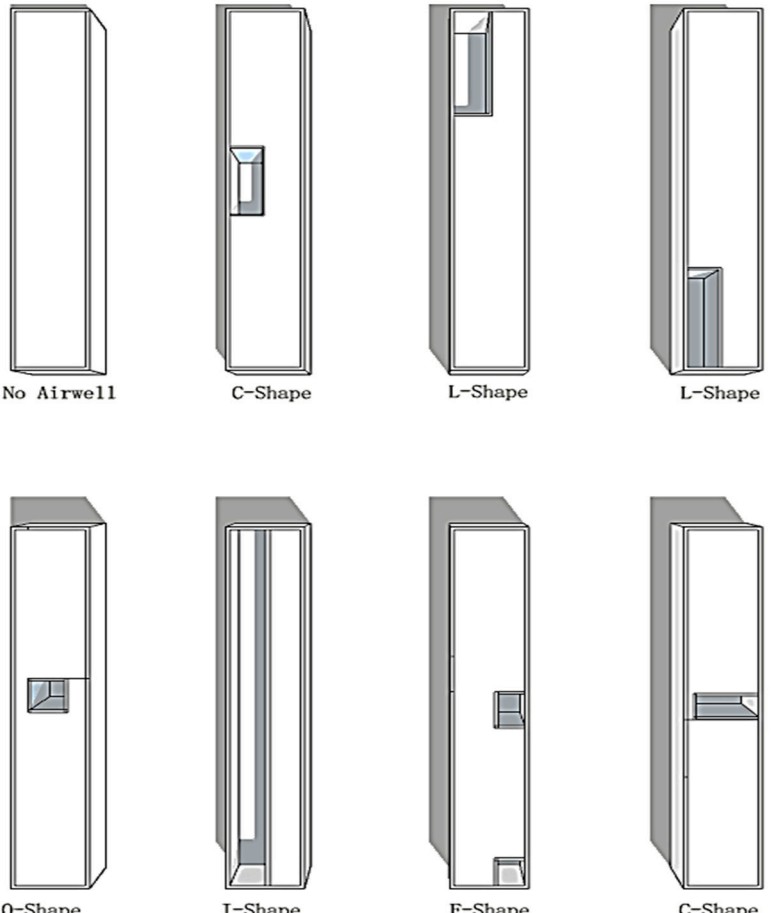

**Figure 1.** Possible positions of an air well in a single-story terrace house [31].

In general, the UBBL designates minimum requirements for openings in walls to induce natural ventilation and daylight. The purpose of the opening requirements is to enhance natural ventilation in enclosed spaces and ensure that windows or openings are able to protect the indoor occupants from direct exposure to rain and the sun. The requirements stated in the UBBL are not specified in terms of dimension, design, or position, thus allowing a certain degree of architectural design flexibility and creativity. Other than the UBBL, the Malaysian Standard MS1525:2007 [33]—Code of Practice on Energy Efficiency and Use of Renewable Energy for Non-Residential Buildings and the Malaysian Standard MS2680:2017 [34]—Code of Practice on Energy Efficiency and Use of Renewable Energy for Residential Buildings have further specified guidelines with respect to thermal and visual comfort, building design, and energy efficiency. Ultimately, the standards could be complemented with the UBBL provisions, which are crucial to be used as a design reference for architects.

### 2.2. Natural Ventilation Strategies via an Air Shaft

In general, there are two types of ventilation strategies, namely, cross ventilation and single-sided ventilation [11,35]. Mohit and Mahfoud [36], via a survey, found out that especially double-story terraced houses could not satisfy occupants' needs for thermal and natural lighting comfort. Thus, most of the occupants opt for mechanical ventilation systems to solve the ventilation problem [7]. Meanwhile, based on Nugroho study [18],

preliminary study using a field measurement on thermal comfort in a single-story terraced house in the context of Malaysia, it is revealed that the design of the single-story house in Malaysia is insufficient and not effective in providing thermal comfort through natural ventilation. This is due to the fact that the single-sided ventilation system in the test room (master bedroom of the terraced house) reduces its ventilation effectiveness. By virtue of this, it is crucial to propose a solar chimney or air well to induce and enhance natural ventilation.

The effectiveness of a solar chimney as a passive cooling tool to improve the indoor thermal conditions in hot, humid climates has been proven in several studies, including the ones conducted in a tropical climate [2,20,37]. Findings show that the application of a ventilation shaft in buildings could enhance natural ventilation. For example, based on an experimental study conducted on a single-story terraced house in Malang, Indonesia, using a solar chimney cum vertical landscape, it was found that the mean air temperature of the indoor environment was within the acceptable comfort range. A combination of both a solar chimney and a vertical landscape could reduce the use of a mechanical ventilation system in domestic buildings and simultaneously provide a natural passive cooling effect to the indoor environment.

There are several ways to address the problem of thermal comfort in terraced houses, and one of them is replacing an air well (with bigger shaft geometry) with a solar chimney (with smaller shaft geometry) [31]. In a hot, humid climate, a solar chimney can provide both thermal comfort and natural daylight for a terraced house. As for the purpose of better thermal performance, a solar chimney, widely applied as a natural ventilation tool, can help induce the air movement, via air-pressure and air-temperature gradient from the inlet of a building, and pass through the occupied zone. The cool air will replace the hot air via air convection, and the hot air will be released through the solar chimney outlet due to temperature differences [37]. A solar chimney is basically composed of glass, enclosed extruded concrete walls, an absorber, and an air gap. The concept of applying solar heat gain to generate passive cooling has been receiving more attention by designers and researchers, especially in hot climate regions. There have been various studies attempted to promote and improve a solar chimney's thermal performance, in which they specify its application based on climatic conditions, building types, and other controlling factors such as inlet and outlet configurations, materials, room depth, and forms.

Wei et al. [38] studied a series of connected solar chimneys on roofs with inclination angles as well as a vertical section facing the south wall. In their findings, the optimal ratio of length to width was found to be 12:1, and the optimal inclination angle to be 4° via mathematical modelling. In another case, Amori and Mohammed [39] investigated the effects of integrating the phase change material (paraffin) with a solar chimney on its thermal behavior. Computational fluid dynamic (CFD) analysis was used to predict thermal performance as well as the two-dimensional fluid flow. The findings show that the phase change material extended the ventilation hours at nighttime by discharging the stored energy from 13:00 h to 22:00 h for 9 h. In addition, in one of the old studies, N.K. Bansal et al. [40] developed a numerical model consisting of two main variables, i.e., sizes of the openings of the solar chimney and the values of the discharge coefficient, for a solar chimney in order to enhance the effect of thermally induced ventilation in buildings.

The findings show that the air velocity of 140 m$^3$/h to 330 m$^3$/h was induced by the solar radiation of 200 W/m$^2$ and 1000 W/m$^2$, respectively, with 2.25 m$^2$ solar collector areas in the solar chimney. Moreover, based on the fundamental numerical model incorporating height and diameter variables, the optimum solar chimney configurations for the case of Tehran should be as follows: a collector inlet of 6 cm, a solar chimney height of 3 m, and a solar chimney diameter of 10 cm, where the velocity of air could speed up to 4 to 25% in different cases [41]. Based on Mathur et al.'s [14] results considering a solar chimney's depth and the inlet height as key parameters, the air change rate is found to increase with the depth of the solar chimney, and was in direct proportion to the solar irradiance. In the

context of a high-rise building, a thermal comfort study was carried out by experimenting with a combination of both a solar chimney and a wetted roof [42].

The literature above on the effect of a solar chimney or air shaft on thermal comfort, though mostly experimental (based on simulations and prototypes) and conducted in the settings of multiple-story buildings, are adequately valid for us to conclude that an air-well system is vital to function as an effective natural ventilation tool, especially with the right combinations of configurations in a building. Therefore, to fill in the existing knowledge gaps, an empirical field measurement study is necessary in this paper.

## 3. Methodology

This methodology section mainly discusses data collection and analysis techniques and procedures involved in the field measurement, which was carried out in the single-story air-welled house located in Kuching, Sarawak, Malaysia. However, prior to that, the next subsection provides justifications of the how the case study of a single-story terraced house with an air well was selected.

*Selection of Case Study House*

The selection of the terraced house type as a case study was based on the terraced house classification data. The classification, presenting four types of single-story terraced house and five types of double-story terraced house layout plans, is provided in Figure 2 [43]. The classification samples, based on 219 floor plans focusing on the internal layout and total floor area (gross floor area), cover the earliest modern terraced houses as well as new terraced houses from 2012–2016. From the classification chart (Figure 2), the case study house was classified as having the common typological layout of terraced houses with total floor area ranging from 85 to 90 m². Having said that, although some variations of layouts may be observed, we believe that they may not significantly influence an air well's thermal performance and thus that the selected terraced house could be fairly represented as a typical case study building for a stack ventilation study [20].

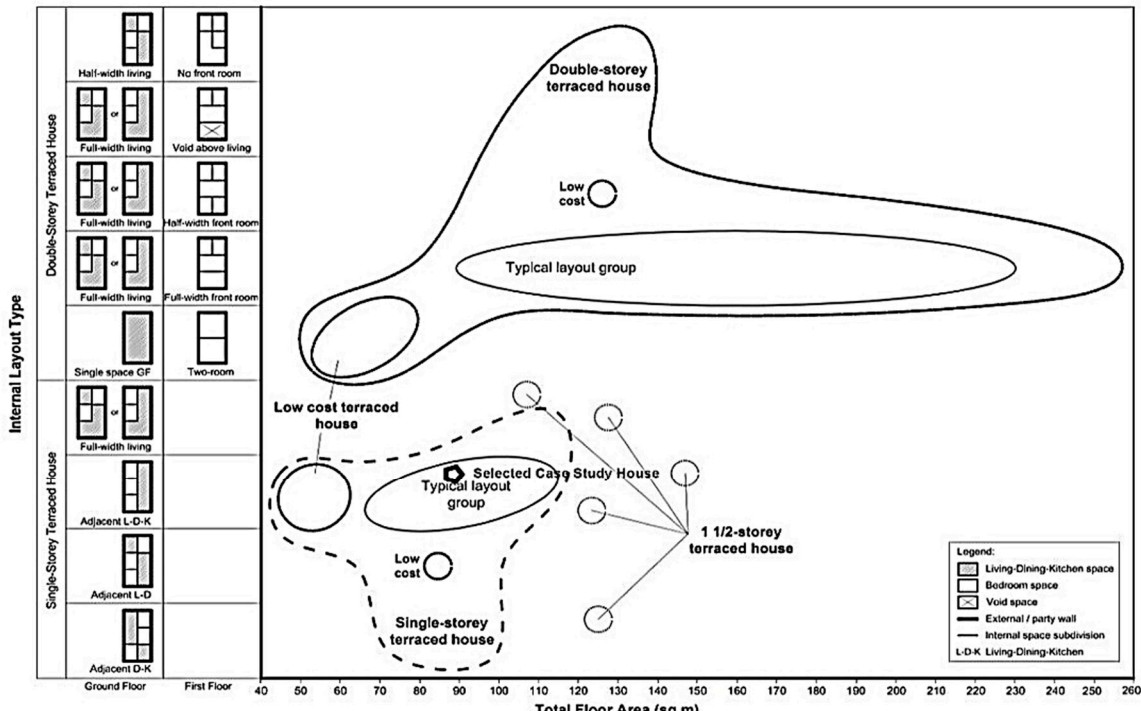

**Figure 2.** Classification of typical Malaysian terraced houses by internal layouts [43].

Since the study is to understand the thermal performance of the existing room with single-sided ventilation, it is a 1:1 full-scale model. To perform this field measurement, a terraced house located 5.65 km to the west of the city of Kuching, Sarawak, East Malaysia, was selected as a case study (see Figure 3). In short, the field measurement was carried out to understand the following existing conditions of the typical single-story terraced house in terms of:

- Outdoor weather conditions in Kuching, Malaysia;
- Thermal performance of the test room with a single-sided opening;
- Thermal performance of the test room attached to the existing air well; and
- Thermal performance of the air well.

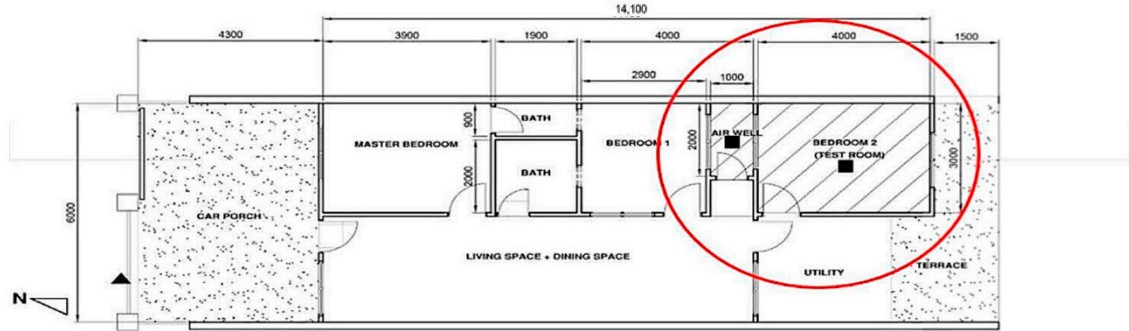

(a) The research study focuses on the bedroom and airwell of case study house (as in red circle). Diagram (a) indicates the floor plan of the case study house and location of the case study room and airwell. Room 2 was selected as case study room since it is attached to the air well and thus the thermal performance of air well could be examined

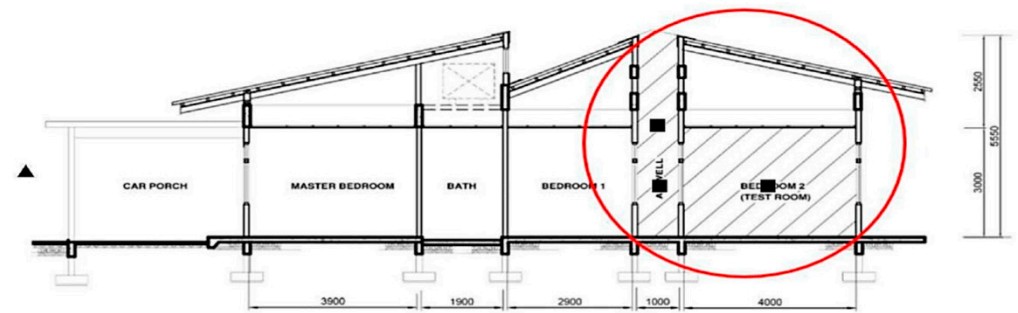

(b) The diagram shows the section drawing of the case study house. The hatches area and red circle indicates the position of case study room and air well. The core unit has been selected for the study instead of whole house since the performance of core unit gives precursors results to the extensive study in future.

**Figure 3.** Field measurement/case study house floor plan (**a**) and section (**b**). The hatched areas in the red circle indicate the focus area in this study. The ▲ in diagram (**a**) represents the position of the outdoor weather station (2.0 m from the floor level and 4.3 m from the external wall of the case study house), and the ■ represents the position of indoor field measurement instruments (1.5 m from the floor level for the test room and the lower air well, and 3.0 m from the floor for the upper air well).

Figure 3a,b illustrate the case study house, where the house frontage faced north, sandwiched by two party walls on both sides. The total walled-up floor area of the house was 84.6 m², with an elongated length of 14.1 m and width of 6 m. There were three bedrooms in the house. Bedroom 1 and Bedroom 2 were attached to the air well, while the master bedroom was a single-sided opening room facing north. In this scenario, Bedroom 2, with a total floor area of 12 m² and a ceiling height of 3 m, was selected as the test room since its end wall was attached to the air well, and the opposite wall openings faced the external environment. The study focused on the test room and the air well only (see Figure 3).

The field measurement was carried out from 3 January to 29 June 2014 (178 days) at a single-story terraced house in Kuching, Sarawak, Malaysia. Although the dataset seems to be rather old, the validity of the data collected can be justified in Figure 4, summarizing the variation of 10 years of climatic data in Kuching, Sarawak. In short, the measured data were valid as the deviation range of the air temperature was deemed insignificant and was quite consistent during the period. Literature has supported that, where there was a very slight variation of average air temperature within the 10-year period, with an increment of 0.39 °C from 2005 to 2015 [44]. Based on the 30 years (2005–2040) forecast study, recorded by the Malaysia Meteorological Centre, the air temperature rises linearly within the range of 26.00 °C to 28.50 °C. According to the 10 years of climatic data from Kuching, Sarawak, from 2006 to 2019, the average annual outdoor air temperature ranged from 26.20 °C to 27.30 °C, whereas the annual mean maximum air temperature ranged from 31.30 °C to 32.5 °C. At the same time, the annual mean minimum air temperature ranged from 22.9 °C to 23.8 °C, and the average wind speed ranged from 5.90 m/s to 7.00 m/s. The differences between the maximum and the minimum for the average annual temperature from 2006 to 2019 was 4.03 °C, whereas the differences between the maximum and the minimum for the annual average maximum temperature throughout the years was 3.69 °C (see Figure 5).

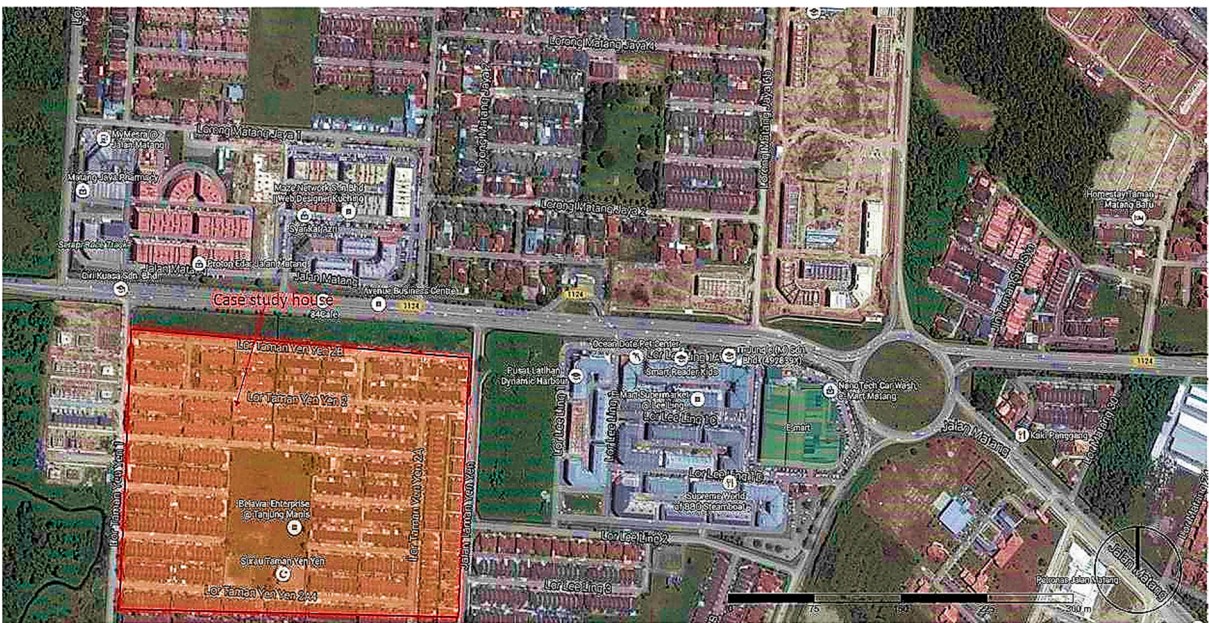

**Figure 4.** The highlighted area indicates the location of the case study house—Taman Yen Yen, Jalan Matang, Kuching, which is 5.65 km away from Kuching city (Image courtesy of Google Maps).

After justifying the validity of the segmented data used, next, descriptions of the case study are provided. That is, a 3.0 m × 3.0 m bedroom, with a floor-to-ceiling height of 3.0 m, consisting of external windows and opposite wall attached to an air well was installed with experimental instruments. The measured data include air temperature and humidity. All measurements were auto-logged at 15 min intervals. The measurement was carried out in a free-running non-air-conditioned test room. The measured test room was not occupied by any occupant, and the operable window was open fully 24 h for a week. During the field measurement, the test room door was closed all the time. The measurement instruments used in this study are summarized in Table 1.

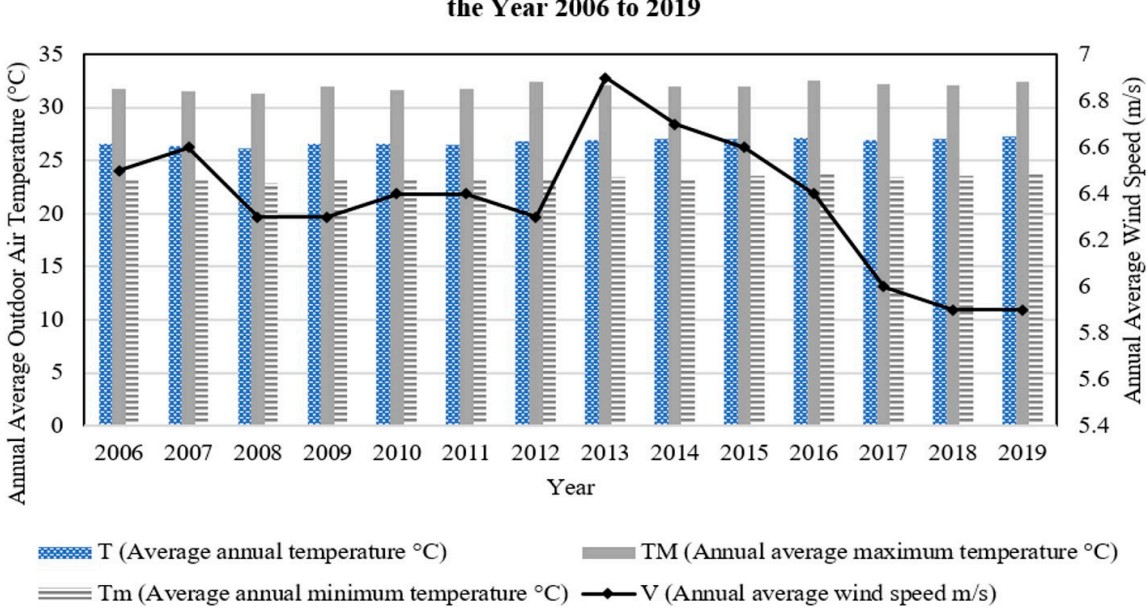

**Figure 5.** Historical data from 2006 to 2019 on annual average outdoor air temperature and wind speed in Kuching.

**Table 1.** Purposes of field measurement instruments used.

| Space | Data Type | Equipment | Setting Positions of Equipment | Descriptions |
|---|---|---|---|---|
| Bedroom 2 (test room) | Air temperature Relative humidity Globe temperature Air velocity, Predicted Mean Vote (PMV) and Predicted Percentage of Dissatisfied (PPD) | Delta Ohm HD32.3 | 1.50 m from the floor level, in the center of the room | Purpose: to understand the current thermal performance of residential rooms in Kuching, Malaysia |
| Air well | Air temperature Relative humidity | HOBOware U12 air temperature and relative humidity data logger, and HOBOware U12 air velocity data logger | Upper air well: 3.0 m from the floor level, lower air well: 1.5 m from the floor level | Purpose: to investigate the thermal performance of the upper and lower air well |
| Outdoor | Air temperature Relative humidity Solar radiation Wind velocity Wind direction | HOBOware U30 weather station | 2.0 m from the floor level | Purpose: The results of the measurements were taken as a controlling factor and compared with the thermal performance of the indoor environment |

## 4. Results and Discussion

Generally, this section discusses the overall field measurement results, including three primary components, namely, (i) outdoor microclimatic weather analysis, and (ii) the thermal performance of the test room and (iii) the air well. For the outdoor climatic analysis, results of selected days (i.e., the hottest and coldest days throughout the period) are reported, in which the measured outdoor data were compared with the meteorological data. The purpose of such a comparison is to verify the trend of air temperatures and

humidity during normal days, hot days, and cold days. Meanwhile, the former were also compared with the indoor measured data comprising the air temperatures and humidity of the test room and the air well.

*4.1. Selected Hot Days (10 to 24 June 2014)*

Selected hot days throughout the measurement period were chosen based on the maximum mean air temperature. According to the dataset, 10 to 24 June 2014, marked the hottest week throughout the second quarter of 2014. The total measurement results ranged from 23.94 °C to 42.99 °C. The average air temperature was 27.88 °C according to the measured data; hence, the selected week was above the average air temperature. The hot days occurred around May to June, due to the Southwest Monsoon season. The hottest air temperature obtained throughout the field measurement was 42.99 °C on 19 June at 17:00. The mean air temperature for the selected days ranged from 27.81 °C on 13 June to 33.55 °C on 19 June 2014. The average deviation of the mean air temperature between the field measurement and meteorological data for the selected days was 2.07 °C.

According to the meteorological data and measured data during the hot period (10 June to 24 June 2014), the minimum temperature difference was 0.51 °C on 13 June 2014, whereas the maximum temperature deviation was 3.14 °C on 19 June 2014. The minimum difference of 1.84% and the maximum difference of 9.35% between both measured and meteorological data show that the differences were insignificant for the extreme hot days. On the other hand, the average relative humidity for the selected hot period was marked as 91.41%. The relative humidity was inversely proportional to the air temperature, meaning that the highest mean air temperature day (19 June 2014) was recorded as the lowest mean relative humidity day with a value of 66.24%, whereas the lowest mean air temperature day (13 June 2014) was marked as the highest mean relative humidity day with a value of 86.34% throughout the selected hot period. In order to understand the temperature variation of the hot day, 19 June 2014, was selected to be analyzed because it was the hottest day during the measured period.

According to Figure 6, the selected hot day was the hottest day throughout the study period with the highest air temperature of 42.99 °C at 5:00 p.m. Even though the temperature difference between the two sets of data at the hottest hour was recorded as 24%, which was the highest deviation, the variation pattern for both sets of data fluctuated at a similar rate. At 5:00 p.m., the meteorological data of the air temperature were recorded as 32.7 °C, which was one of the highest air temperatures throughout the day. The deviation of air temperature ranging from 0.115 °C at 3:00 a.m. to 10.292 °C at 5:00 p.m. was 10.117 °C different between the maximum and minimum deviation values. This indicates that the existence of variations is accepted since the fluctuation patterns were within the same direction and range. In the relative humidity context, the overall mean relative humidity of the selected day was 66.24%, with the highest value recorded as 90.8% at 6:00 a.m. and the lowest value 35.75% at 7:00 p.m. The impact of a hot day would directly cause thermal discomfort to the occupants since the heat gain from the outdoor environment was transmitted via radiation, convection, and conduction of the air and building material to the occupants. The analysis of the hot day could predict the air temperature for the indoor environment and refine the solution to reduce the thermal discomfort of the occupants.

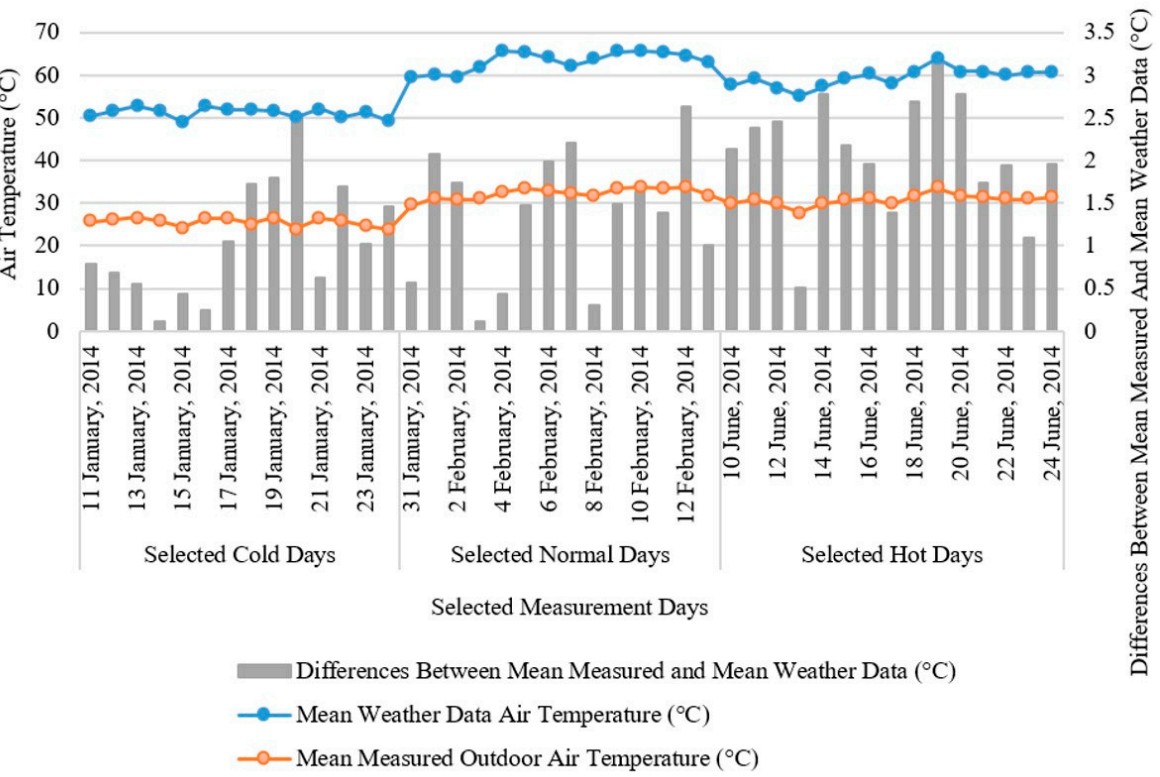

**Figure 6.** Overall comparison between field measurement and meteorological data for selected measured cold, normal, and hot days throughout the study period.

### 4.2. Selected Cold Days (11 January to 24 January 2014)

After analyzing the selected hot day condition for the case study house, an analysis for the cold days is important to be done to understand the microclimate for the case study. The cold days happened between the end of December and early February, which was the peak season of the Northeast Monsoon that tends to cause a heavy rainfall. Figure 6 shows that from 11 January to 24 January 2014, the field-measured mean air temperature ranged from 23.94 °C, which happened on 24 January 2014, to 26.66 °C on 19 January 2014. Throughout the field measurement period from 3 January to 29 June 2014, the lowest mean air temperature happened on 5 February 2014, with 19.57 °C, and the most frequently occurring mean air temperature below the overall mean air temperature was 24.73 °C. Hence, the selected period to represent cold days possessed a mean air temperature ranging from 23.94 °C on 24 January to 26.66 °C on 19 January 2014.

The average deviation of the field measurement data and weather data was 1.02 °C, which is considered acceptable and insignificant. The highest deviation mean value throughout the selected cold days was 2.15 °C on 20 January 2014, where the weather station showed 26.17 °C and the field measurement was 24.01 °C. The massive and dynamic atmospheric circulation occurring in the spacious flat topography caused the weather station data to have a lower air temperature compared to the case study field measurement. The lowest deviation value was 0.11 °C (which was 0.43%) on 14 January 2014, with a field measurement value of 25.73 °C and weather data of 25.85 °C.

The mean relative humidity for the selected cold days ranged from 80.31% on 16 and 17 January 2014 to 95.57% on 15 January 2014. Similar to the hot days' condition, the day with a higher air temperature was accompanied with a lower relative humidity, while the day with the lower air temperature came with a higher relative humidity. The selected cold

days with the highest relative humidity, which was 95.57%, happened on 15 January 2014 with a mean air temperature of 24.27 °C. On the other hand, the cold days with the lowest relative humidity, which was 80.31% on 16 and 17 January 2014 had a mean air temperature of 26.53 °C. The fluctuation of relative humidity in cold days was lower compared to the hot days. The relative humidity ranged from 66.24% to 86.38% and the cold days from 80.31% to 95.57%. This shows that relative humidity for cold days was not only high but also stable.

When the study specified the selected cold day, as shown in Figure 6, the fluctuation pattern of the graph shows a more dynamic pattern compared to the selected days graph as shown in Figure 6. The selected cold day is represented by 24 January 2014. In general, the measured data were lower than the weather station air-temperature value. The differences of both sets of data were recorded as 2.21 °C or equivalent to 8.87%. The measured mean air temperature ranged from 22.681 °C at 7:00 a.m. and 25.635 °C at 6:00 p.m., whereas the weather station data air temperature ranged from 23 °C at 5:00 a.m., 6:00 a.m., and 7:00 a.m. to 30 °C at 2:00 p.m., 3:00 p.m., and 4:00 p.m. The data from the weather station were more general compared to the field measurement data. The percentage differences between both sets of data ranged from 1% to 22%. The deviation gaps were larger when the air temperature increased, which was in the afternoon. The field measurement data show a more stable and consistent pattern whereas the weather station data possess more significant fluctuation range. The field measurement data were recorded based on microclimate condition, since the field measurement instruments were located at human-level height.

Furthermore, relative humidity for the cold day recorded was 60.44% at 2:00 p.m. and 95.96% at 7:00 a.m. The relative humidity from 12:00 a.m. to 8:00 a.m. on 24 January 2014, was above 89% but it gradually dropped after 9:00 a.m. until 8:00 p.m. When night fell, the relative humidity increased from 72.99% at 7:00 p.m., to 78.52% at 8:00 p.m., to 81.34% at 9:00 p.m., to 82.75% at 10:00 p.m., and to 83.89% at 11:00 p.m. Diurnal air temperature and diurnal relative humidity maintained the high temperature–low humidity relationship even though it was categorized as a cold day. The impact of a cold day to the indoor environment was not as critical as that of a hot day since thermal performance of hot days would cause significant impact on the thermal comfort of occupants in the tropics.

*4.3. Selected Normal Days (31 January to 13 February)*

After looking into selected hot and cold days from the field measurement days, it is important to look into outdoor air temperature and relative humidity of normal days, which falls on the total average of air temperature throughout the field measurement. The total average air temperature of the field measurement was 27.88 °C. The selected normal days were from the end of January to around the end of April, during the inter-monsoon period.

Figures 6 and 7 show the variation and deviation of air temperature between the field measurement and weather station for normal days. In general, the percentage deviation between both sets of data was acceptable since the highest percentage of deviation throughout the selected days was not more than 7%. The mean air temperature of the selected normal days obtained from the field measurement ranged from 25.34 °C on 31 January to 28.53 °C on 12 February 2014, whereas the data from the meteorological center ranged from 25.38 °C to 28.04 °C on 3 February and 11 February 2014, respectively. The average deviation for both sets of data was 0.64 °C, which was not significant or acceptable. During the normal days, the highest deviation percentage between both sets of data was found on 13 February 2014, with a deviation of 6.36%, whereas the closest value was found on 4 February 2014, with a deviation of 0.03%. However, the highest percentage of deviation on 13 February was only 1.67 °C, whereas the lowest percentage deviation was 0.007 °C on 4 February 2014. It can be inferred that the measured data are reliable and similar to the weather station data.

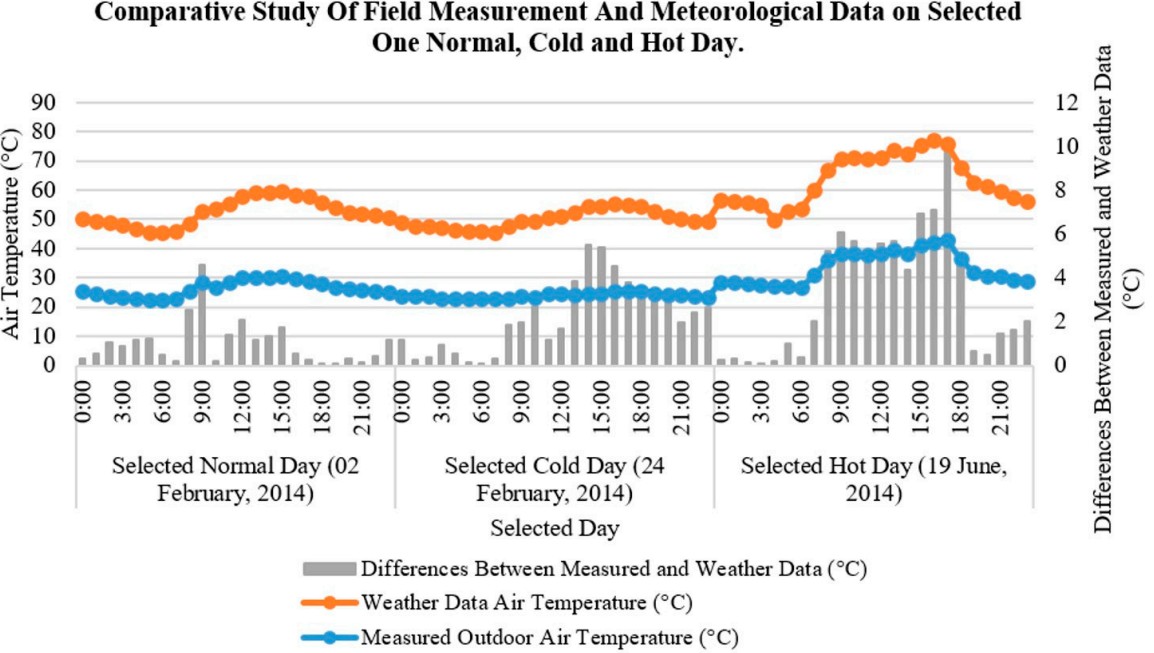

**Figure 7.** Comparison between the field measurement and meteorological data for one selected normal, cold, and hot day throughout the study period.

For the relative humidity, the highest relative humidity and lowest relative humidity values were 81.53% and 67.28%, respectively. The highest relative humidity happened on 31 January and the lowest on 5 February 2014. The average relative humidity from 31 January to 13 February was recorded as 77.27%. The mean relative humidity for selected normal days was similar to the range suggested by Malaysia Meteorological Department [45].

Figure 7 also shows that the selected day to study the daily air temperature and relative humidity variation was 2 February 2014. The variation order for both meteorological data and field measurement data agreed with each other and had a similar pattern except for 8:00 a.m. and 9:00 a.m., which showed 10% and 16%, respectively. Out of 24 h of the selected normal day, each of the deviations was not more than 8% except for 8:00 a.m. and 9:00 a.m. The field measurement data at 8:00 a.m. and 9:00 a.m. showed 25.51 °C and 28.61 °C, whereas the weather data showed 23 °C and 24 °C. The deviation happened around sunrise when radiation started to warm up the air. The weather station was located on a flat terrain where the water vapor density was higher, which caused the air temperature to be lower. The mean air temperature for the field measurement ranged from 22.52 °C at 6:00 a.m. to 30.74 °C at 3:00 p.m. Since 2 February 2014, was selected to represent a normal day, the mean air temperature range could indicate that before sunrise and after sunset (nocturnal), the case study house experienced lower air temperature as well as a high mean daytime (diurnal) air temperature. This means that daytime thermal performance is more critical compared to nighttime thermal performance.

Relative humidity for a normal day could range from 63.24% to 93.18%. The lower relative humidity happened at the hottest hour, which was 3:00 p.m. whereas the highest relative humidity happened at one of the coldest hours, which was 7:00 a.m. A difference of 29.94% between highest and lowest air humidity shows that occupants in tropical climates experience dynamic changes of thermal comfort within 24 h, from daytime to nighttime, throughout the year. This demonstrates that thermal comfort could be an issue for a free-running building, especially during daytime, since the climate is hot and humid throughout the year.

In the next section, an analysis of outdoor air temperatures is detailed through the discussion on the daily maximum and the daily mean for air temperatures, relative humidity, solar radiation, wind velocity, and wind directions.

### 4.4. Field Study Results: Outdoor Climate

This section examines the four different main climatic parameters, namely, air temperature, solar radiation, relative humidity, and wind velocity. In this case, the daily maximum temperature was discussed, since the maximum daily air temperature is important to be taken as the worst-case scenario for the future study, e.g., a boundary condition for modeling simulation. In the hot, humid, tropical climate, high temperatures for both outdoor and indoor environments cause concerns because they lead to thermal discomfort. The poor thermal performance of a building increases the tendency of the occupants to use mechanical ventilation systems.

#### 4.4.1. Daily Maximum

The daily maximum of the parameters is considered to be more important compared to the daily minimum. Under the hot and humid tropical climate condition, extreme and high air temperature, relative humidity, solar radiation, high intensity of outdoor wind during a certain period of time—especially monsoon seasons and so forth—could directly and indirectly affect indoor thermal performance. Hence, this subsection discusses the daily maximum of the following selected parameters.

#### 4.4.2. Air Temperature

Figure 8 shows the daily maximum for outdoor air temperature throughout the field measurement days. The maximum air temperature of each day was recorded in order to investigate the extreme condition of the microclimate throughout the field measurement. The overall daily maximum air temperature ranged from 25.35 °C to 42.99 °C on 20 January and 19 June, respectively. The air temperature difference of 41.11% between the maximum and minimum values shows that the microclimate changed significantly according to the monsoon seasons. The Northeast Monsoon, which usually happens around January, causes a high volume of rainfall and directly lowers the air temperature. According to McGinley [46], Sarawak receives minimal rainfall in June and July annually. Without adequate rainfall, the phenomenon directly causes drought around the coast due to the high intensity of solar radiation and hot air convection. Therefore, June and July are considered critical months for thermal comfort. Furthermore, haze pollution, which happens from June onwards, would be one of the reasons causing extreme high temperatures around mid-year [47].

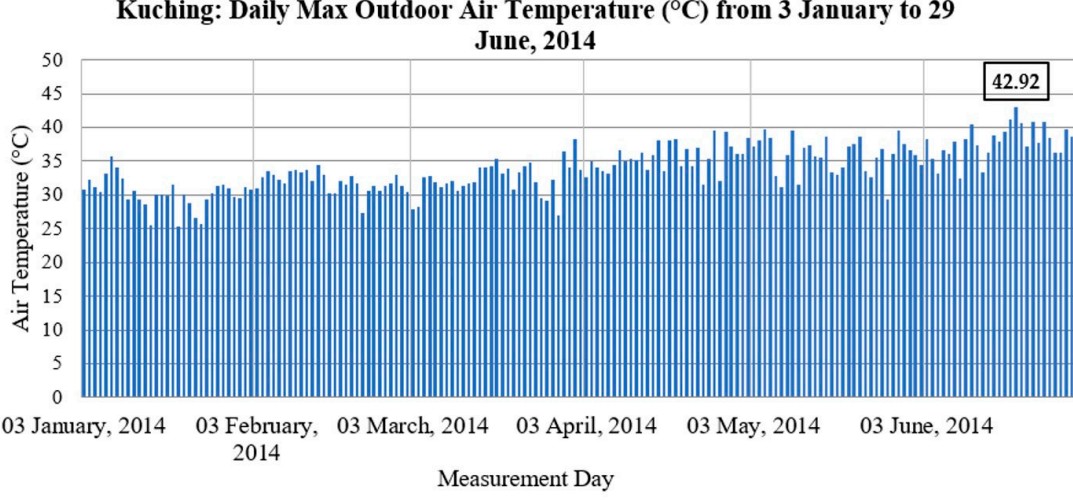

**Figure 8.** Daily maximum outdoor air temperature throughout the field measurement (3 January to 29 June 2014).

### 4.4.3. Solar Radiation

The daily maximum of solar radiation for Kuching throughout the field measurement ranged from 86 W/m$^2$ at 6:00 p.m. on 27 May 2014 to 1022 W/m$^2$ at 1:00 p.m. on 3 March 2014. The solar radiation directly influenced the air temperature through conduction and air convection. The air temperature for the studied day could be deduced from the daily maximum solar radiation. According to Makowski [48], since the solar flux only influences the diurnal daylight, it significantly affects the daily maximum air temperature compared to the daily minimum air temperature, whereas the nocturnal air-temperature variation is affected by thermal radiative exchanges. Nighttime surface air radiative cooling relies on the atmospheric capacity to absorb and conduct the thermal radiation towards the Earth's surface. In this case, the daily maximum study was significant in understanding the effect of diurnal conditions, especially daily maximum air temperature and solar radiation, in order to deduce the extreme thermal conditions for an indoor environment.

Figure 9 indicates the daily maximum solar radiation throughout the field measurement. Comparable to Figure 8, the highest daily maximum air temperature was 42.99 °C on 19 June, whereas the highest daily maximum for solar radiation was 1022 W/m$^2$ on 3 March 2014. The maximum air temperature for 3 March 2014 was 27.88 °C even though it had the highest daily maximum solar radiation. In general, high solar radiation is accompanied by high air temperature.

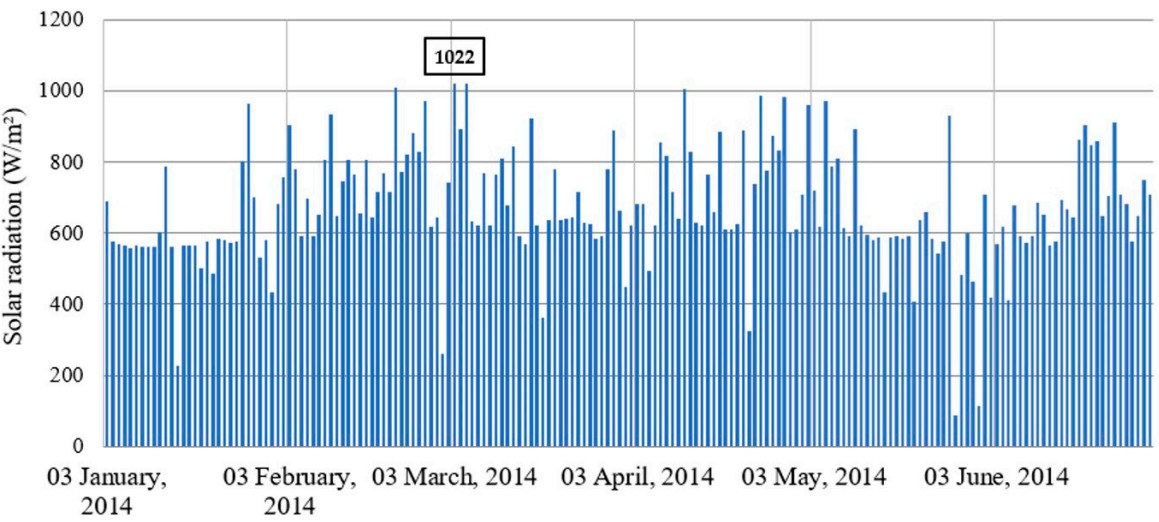

**Figure 9.** Daily max solar radiation throughout the field measurement (3 January to 29 June 2014).

This rare case could be due to other factors such as high cloud cover rate, high evaporation rate on the measuring area, high wind velocity, and others. According to Graham [49], low and thick clouds (example: stratocumulus) could reduce the Earth's surface and air temperature since it reflects the solar radiation, whereas high and thin clouds (example: cirrus clouds) allows direct sunlight to penetrate, which causes high air temperature. The high and thin clouds also trap heat and infra-red radiated from the Earth's surface.

Other than that, clear moving air, humidity, and clouds also balance the air temperature on Earth. In order to explain the condition that high solar radiation could slightly increase the air temperature, three randomly selected days with high solar radiation throughout the field measurement are discussed briefly. The chosen dates are 5 March 2014,

with an air temperature of 30.91 °C at 1:00 p.m.; 28 April 2014, with an air temperature of 36.17 °C at 1:00 p.m.; and 18 June 2014, with an air temperature of 41.19 °C at 2:00 p.m. The solar radiation value was 1019 W/m², 981 W/m², and 903 W/m², respectively. The dates above were selected due to the high solar radiation value throughout the field measurement period. From the selected field measurement results, it can be deduced that high solar radiation causes high air temperature in general, due to the high reflectivity rate of emitted radiation from solar radiation to the Earth's surface.

### 4.4.4. Wind Velocity

Wind velocity near to the Earth's surface is one of the most influential thermal comfort parameters for the indoor environment. In order to understand the indoor thermal performance, the intensity and speed of outdoor wind velocity must be determined. According to the Beaufort scale of wind speed, a wind speed of more than 5.4 m/s could lead to uncomfortable conditions for the occupants. The recommended comfortable range in a tropical context, based on the Beaufort scale, ranges from 1.6 to 5.4 m/s [50]. Figure 10 depicts the daily maximum air velocity throughout the field measurement. Out of 4270 data for each of the hours within the field measurement periods, 2641 data ranged from 1.6 m/s to 5.4 m/s at daily maximum conditions. The highest daily maximum wind velocity was marked as 8.334 m/s on 19 May 2014, whereas for the normal hour it could be as low as 0 m/s. The extremely huge range of wind velocity is a critical dilemma for the indoor environment since it cannot give a consistent cooling effect to the indoor environment. Furthermore, the poor layout design of modern terraced housing hardly creates a pressure gradient between the indoor and outdoor environment to induce wind ventilation.

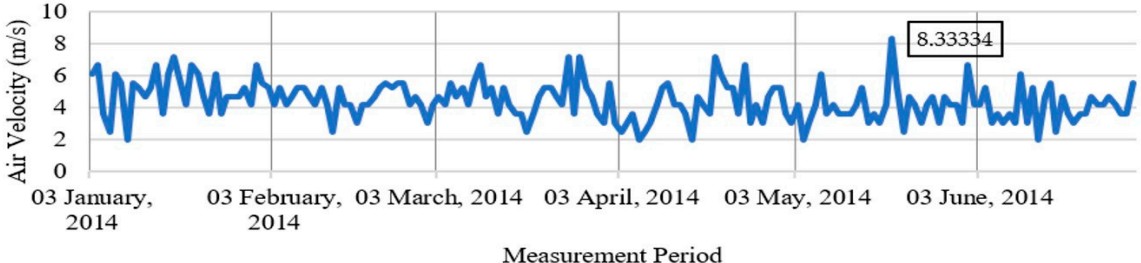

**Figure 10.** Daily maximum air velocity throughout the field measurement period (3 January to 29 June 2014).

The daily maximum wind velocity in Figure 10 shows the daily maximum air velocity of each day throughout the field experiment. The highest daily maximum and lowest daily maximum wind velocity ranged from 1.95 m/s on 13 June to 8.34 m/s on 19 May 2014. The highest daily maximum wind velocity happened during the monsoon season, whereas the lowest daily maximum wind velocity occurred during the end of inter-monsoon season. The effect of monsoon wind directly influences the climate in general.

### 4.4.5. Relative Humidity

The other important parameter that affects thermal comfort in the tropics is relative humidity. Other than high air temperature throughout the year, high relative humidity is also one of the critical factors that contributes to thermal discomfort. According to Figure 11, daily maximum relative humidity obtained throughout the field measurement ranged from to 82.89% to 100%, which happened on 20 June, as well as 6 January and 29 June. High relative humidity usually happens during the lowest air temperature, which is during rainy days or nighttime.

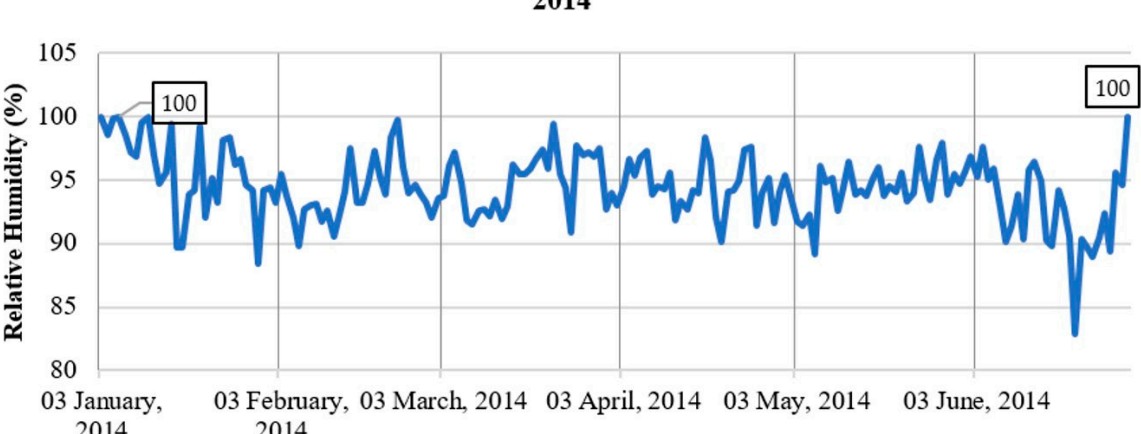

**Figure 11.** Daily maximum relative humidity throughout the field measurement period (3 January to 29 June 2014).

The lowest daily maximum relative humidity is considered above average or high value. Sahabuddin [50] stated that the ideal relative humidity range for indoor thermal comfort in tropical climates ranges from 30% to 60%. In comparing the outdoor lowest daily maximum relative humidity, the critical number could be one of the factors that needs to be noted while considering the passive cooling architectural design. However, a natural ventilation method is hardly implemented widely due to the lack of research. Hence, increasing air velocity to reduce air temperature and promote thermal comfort in tropical climates is one of the most reliable solutions under a natural ventilation method [51–53].

In the next section, in order to investigate the relationship between outdoor climatic conditions, thermal performance for an air well, and a single-sided ventilated test room, a comparative analysis was carried out.

*4.5. Field Study Results: Comparison between the Minimum, Mean, and Maximum Thermal Performance between the Air Well, Test Room, and Outdoor Climate Conditions*

Comparison of Daily Maximum Air Temperature and Relative Humidity for Outdoors, Test Room, and Air Well from 3 January to 29 June.

A comparison between the maximum air temperature for three locations, outdoors, the test room, and the air well, justifies the correlation between each other. An understanding of the mutual relationship between outdoors and test room (single-sided ventilation); outdoors and the air well (performance of stack ventilation tool), and the test room and air well would enhance the research study background in improving the thermal performance of a habitable room in a single-story house.

Figures 12 and 13 show daily maximum air temperature and relative humidity for outdoors, the test room, and the air well. Both figures are intended to display the variation of overall thermal performance for outdoors, the air well, and the test room. In the previous section, outdoor weather data taken from the field measurement were discussed. However, the same data were applied to compare with the measured indoor environment as reference.

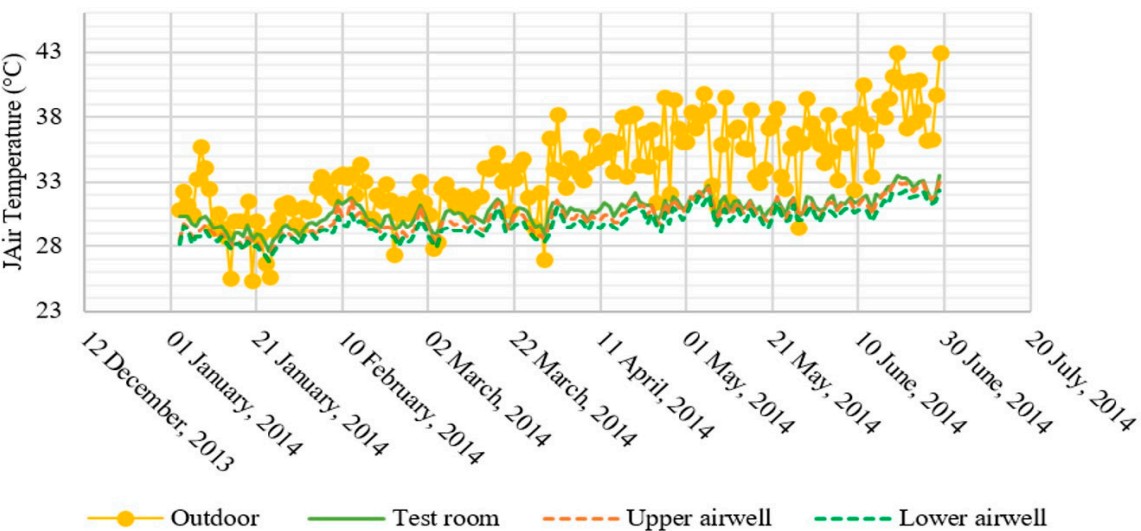

**Figure 12.** Daily maximum air temperature for outdoors, the test room, and the air well throughout the field measurement (3 January to 29 June 2014).

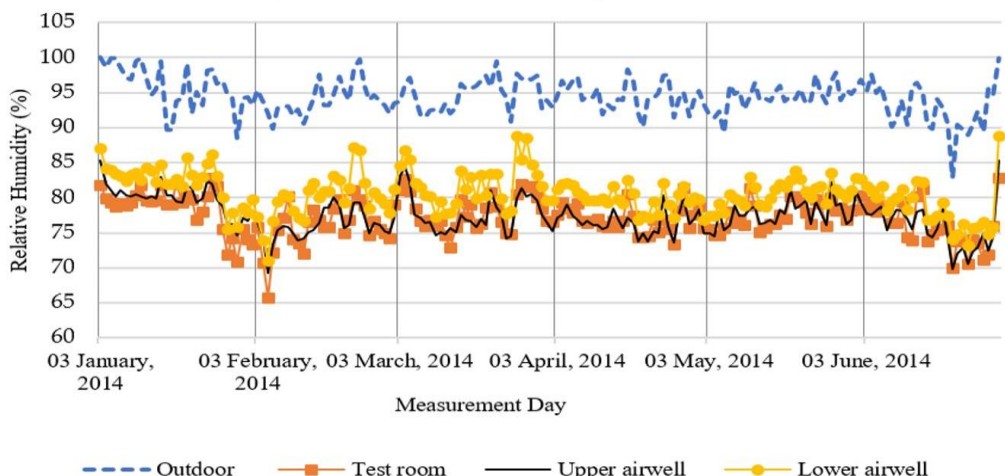

**Figure 13.** Daily maximum relative humidity for outdoors, the test room, and the air well throughout the field measurement (3 January to 29 June 2014).

In general, the daily maximum air temperature for outdoors, the test room, and the air well fluctuated upwards gradually from January to June. Since relative humidity was inversely proportional to air temperature, the overall variation pattern fluctuated downwards gradually. The phenomenon infers that air temperature increased from March. The total average of the daily maximum outdoor air temperature was 3.19 °C and 3.92 °C higher compared to both the test room and air well, respectively. Furthermore, the total average of the daily maximum outdoor relative humidity was 17.26% and 15.47% higher than the test room and air well, respectively. The highest daily maximum air temperature deviations for outdoors–test room and outdoors–air well were 9.52 °C and 10.41 °C, respectively, whereas the lowest daily maximum air temperature deviations were 3.13 °C and 2.71 °C, respectively. The highest deviation happened on the hottest day, which was 19 June, and the lowest deviation occurred on a cold day, which was 20 January 2014. From

the findings, it is evident that thermal performance of the indoor environment was stable since the fluctuation patterns were more consistent.

Based on the above detailed results and findings, the key thermal performance results of the case study house in terms of temperatures of outdoors, the test room, and the air well are summarized in Tables 2–4.

**Table 2.** Summary of field measurement—outdoor weather data for selected days' duration and one selected day (hot, cold, and normal days).

| Condition | Mean Outdoor Air Temperature (°C) | Maximum Outdoor Air Temperature (°C) | Mean Outdoor Relative Humidity (%) | Maximum Outdoor Relative Humidity (%) |
|---|---|---|---|---|
| Hot days (10 June 2014–24 June 2014) | 30.74 | 38.90 | 73.48 | 91.42 |
| Cold days (11 January 2014–24 January 2014) | 25.58 | 28.69 | 86.45 | 95.13 |
| Normal days (31 January 2014–13 February 2014) | 27.95 | 32.01 | 79.87 | 92.25 |
| Hot day (19 June 2014) | 33.55 | 42.99 | 66.24 | 90.56 |
| Cold day (24 January 2014) | 23.94 | 25.63 | 75.59 | 95.95 |
| Normal day (2 February 2014) | 26.42 | 30.74 | 79.64 | 93.18 |

**Table 3.** A summary of the field measurement—thermal performance of the test room for selected days' duration and one selected day (hot, cold, and normal days).

| Condition | Mean Test Room Air Temperature (°C) | Maximum Test Room Air Temperature (°C) | Mean Test Room Relative Humidity (%) | Maximum Test Room Relative Humidity (%) | Average Test Room Mean Radiant Air Temperature (°C) | Maximum Test Room Mean Radiant Air Temperature (°C) | Maximum PMV | Mean PMV | Maximum PPD (%) | Mean PPD (%) |
|---|---|---|---|---|---|---|---|---|---|---|
| Hot days (10 June 2014–24 June 2014) | 31.10 | 32.42 | 70.16 | 75.15 | 30.42 | 31.67 | 2.58 | 2.32 | 94.63 | 87.73 |
| Cold days (11 January 2014–24 January 2014) | 28.04 | 28.84 | 76.00 | 79.79 | 27.33 | 28.09 | 1.80 | 1.64 | 66.96 | 58.55 |
| Normal days (31 January 2014–13 February 2014) | 30.39 | 31.3 | 70.52 | 76.5 | 29.62 | 30.6 | 2.30 | 2.15 | 88.34 | 82.70 |
| Hot day (19 June 2014) | 32.11 | 33.48 | 67.91 | 76.05 | 31.42 | 32.75 | 2.78 | 2.53 | 97.59 | 93.31 |
| Cold day (24 January 2014) | 28.92 | 29.95 | 70.86 | 78.83 | 28.18 | 29.16 | 1.99 | 1.80 | 76.37 | 66.89 |
| Normal day (2 February 2014) | 28.81 | 29.8 | 69.70 | 73.28 | 28.11 | 29.03 | 1.97 | 1.77 | 75.26 | 65.51 |

**Table 4.** A summary of the field measurement—thermal performance of the air well for selected days (hot, cold, and normal days).

| Condition | Mean Air well Air Temperature (°C) | Maximum Air well Air Temperature (°C) | Mean Air Well Relative Humidity (%) | Maximum Air Well Relative Humidity (%) |
|---|---|---|---|---|
| Hot days (10 June 2014–24 June 2014) | 30.92 | 31.75 | 72.67 | 76.26 |
| Cold days (11 January 2014–24 January 2014) | 27.48 | 28.06 | 79.25 | 81.87 |
| Normal days (31 January 2014–13 February 2014) | 29.76 | 30.05 | 73.76 | 78.06 |
| Hot day (19 June 2014) | 31.74 | 32.58 | 71.44 | 75.79 |
| Cold day (24 January 2014) | 28.24 | 28.83 | 75.72 | 79.58 |
| Normal day (2 February 2014) | 28.37 | 28.77 | 74.30 | 78.90 |

## 5. Conclusions

To date, thermal performance of terraced housing is still unresolved, especially in the tropical climate. The limitation of external openings due to the bounded terraced housing layout leads to restrictions in terms of internal spaces designs. These shortcomings have resulted in poor ventilation performance; by which it increases thermal discomfort. This paper studied the thermal performance of a single-story terraced air-welled house via an empirical measurement, where it primarily measured both temperatures and humidity of outdoor weather conditions, a test room (i.e., Bedroom 2), and an air well. Based on the analysis, these are the key findings: (i) The outdoor mean air temperature ranged from 25.58 °C to 30.74 °C, with relative humidity ranging from 73.48% to 86.45%; (ii) the mean air temperature of the air well ranged from 27.48 °C to 30.92 °C whereas the mean relative humidity ranged from 72.67% to 79.25%; and (iii) the mean air temperature for the test room (single-sided ventilation room) ranged from 28.04 °C to 30.92 °C with a relative humidity of 70.16% to 76%; (iv) even though the air temperature of the test room was similar to the outdoor air temperature, due to the high relative humidity, it caused the heat to be trapped in the indoor environment and that led to a static air condition (0.00 m/s); and (v) lastly, based on the aforesaid conditions, the Predicted Mean Vote (PMV) ranged from +1.80–2.58 under the condition of 0.5 clo and 1.0 met, whereas Predicted Percentage of Dissatisfied (PPD) was within the range of 66.96–94.63%. These entail that the current thermal performance of the air well in the terraced house may not work effectively since the positive values of PMV and PPD indicated that the room was hot and thus occupants may have thermally felt discomfort (see ASHRAE 55 and ISO 7730). Besides, regardless of any day (whether it was cold, hot, or normal), the low thermal comfort of the test room was true, as supported by DIN EN 13779 and ISO 7730 (see the Fanger method).

In other words, the existing provision of a minimum of 10% openings of the total floor area of the room for the purpose of natural ventilation, stipulated under the Malaysia Uniform Building By-Law 1984, is proven to be less meaningful because such an imposition does not effectively provide good thermal performance. These empirical findings are of importance, offering novel policy insights and suggestions to the existing building code standard and bylaws. Strict compliance with and necessity for the bylaw requirement should be revisited and further studied. Therefore, further research beyond the recommendation by local regulations in terms of air well or shaft configurations and fenestration geometry (inlet and outlet of openings) could be explored to enhance the ventilation effectiveness (thermal performance) of an air well for the indoor room of a terraced house.

Despite the above contributions and policy implications, this study is not without limitations. The results presented are limited to only one case study (i.e., a single-story house), and the field measurement only lasted for about six months. Although these limitations have been well justified in the Methodology section and have empirically produced valid results, the sample size and time period of the current study can be increased so that a longitudinal comparative study can be conducted in order to provide accurate and more reliable results.

**Author Contributions:** All authors contributed to the manuscript. Conceptualization, M.H.A., D.R.O., and G.H.T.L. E.A. and W.H.C. collected and evaluated, in a coordinated way, all the publications mentioned in the paper. D.N.T. contributed to visualization & editing. With this material, P.C.L. wrote a preliminary version of the article that was further enriched with suggestions and contributions by the rest of the authors. All authors have read and agreed to the published version of the manuscript.

**Funding:** This research publication was funded by UTM Transdisciplinary Research Grant No. Q.J130000.3552.07G55.

**Acknowledgments:** The authors wish to deliver appreciation to Building Science Laboratory, Faculty of Built Environment and Surveying, Universiti Teknologi Malaysia for the field measurement instruments as well as the owners of the case study house—Leng Wee Woon and Liew Thai Jin—for their permission to set up the field measurement for the study.

**Conflicts of Interest:** The authors declare no conflict of interest. The funders had no role in the design of the study; in the collection, analyses, or interpretation of data; in the writing of the manuscript, or in the decision to publish the results.

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
