# Peer review of "Thermal Performance of Single-Story Air-Welled Terraced House in Malaysia: A Field Measurement Approach"

_sustainability, doi:10.3390/su13010201_

Round 1

Reviewer 1 Report

Thanks for addressing most of my comments.

Your response states, "The % deviation for temperature has been removed and changed into the differences of air temperature between measured data and weather data." But some figures (such as Figures 6 & 7) still show a % difference in air temperature. Please make corrections.

Author Response

Comment

  1. Your response states, "The % deviation for temperature has been removed and changed into the differences of air temperature between measured data and weather data." But some figures (such as Figures 6 & 7) still show a % difference in air temperature. Please make corrections

Respond

  1. Thanks for the comments and reminder. In other to avoid confusion we have removed the % deviation of the air temperature especially in Figure 6 and 7. Please see Figure 6 and 7 (line 348 and line 351)

Reviewer 2 Report

The article deals with the evaluation of the effect of the Air-well on the thermal performance of a terrace house in Malaysia through field measurements. The subject deals with a certainly interesting topic.

The paper has been subject to previous revisions and it was now resubmitted. This time the authors followed all the suggestions provided in previous reviews and that effort was appreciated.

It still remains some minor corrections:

  • The subdivision in “Hot days”, “Cold days” and “Normal days” should be defined with more clear intervals, in particular, the +-27.88°C is not a precise definition of the "Standard days".
  • In Table 2, please check the dimensions of the characters.
  • For completeness, please add in the text the values of clothing and metabolic rate used for the determination of the PMV (even if it is ancillary information).
  • Please do not split the tables into two pages.

Author Response

Comment 1

The subdivision in “Hot days”, “Cold days” and “Normal days” should be defined with more clear intervals, in particular, the +-27.88°C is not a precise definition of the "Standard days".In Table 2, please check the dimensions of the characters

Respond 1

Thanks for the comment. In order to avoid confusion, we have removed the text (eg: 27.88°C) at the subdivision’s title. The range of the measured data has added in Line 328-329

The gap of the characters in Table 2 has been changed to normal gap (from “justified” format to “aligned left”, the size of the font has been fixed and the table has been joint and shifted to next page.

Comment 2

For completeness, please add in the text the values of clothing and metabolic rate used for the determination of the PMV (even if it is ancillary information).

Please do not split the tables into two pages.

Respond 2

Thanks for the suggestion. The PMV clothing and metabolic rate value has been added to line 647.

The split table 2 has been combined.

This manuscript is a resubmission of an earlier submission. The following is a list of the peer review reports and author responses from that submission.

Round 1

Reviewer 1 Report

While the study topic is relevant and interesting, the paper needs significant improvement. The paper only presents data for select periods; it would be desirable to present a whole-year data to conclude the effectiveness of natural ventilation performance in terraced housing with an air well system.

The background information and details of the terrace house seem to be too long.

Callings ordinary sensors such as HOBO temperature and humidity sensor as “scientific equipment” sounds like advertising the sensor.

The conclusion section needs to add an objective of the study and some background information before presenting the results.

Presenting % deviation in temperature reading in °C is misleading as °C is not an absolute unit. Therefore, if you change the unit to °F, the % deviation values would also change. Therefore, suggest removing % deviation for temperature data.

Some editorial comments:

A space between a number and unit is missing in many places.

519: “The average deviation of the mean air temperature from field measurement and meteorological data for the selected days is 2.074ºC.” it is presenting too many digits compare. Please report only significant figures. https://en.wikipedia.org/wiki/Significant_figures could be helpful.

590: “The deviation of both set data was 2.21ºC with 8.87%.” It is not clear what 8.87% is referring to.

Use the SI unit consistently. For example, “In Malaysia, a typical single terraced house unit consists of a built-up area of 650-700 square 163 feet with 6.5m of front width, and 11m in length” use SI and IP units mixed up.

Reviewer 2 Report

The article deals with the evaluation of the effect of the Air-well on the thermal performance of a terrace house in Malaysia through field measurements. The subject deals with a certainly interesting topic.
The authors have put great effort into the collection of numerous data both from literature and through an extensive measurement campaign and it is clear the attempt to create a complete paper, however, the article presents some problems described below.

General comments:
- The article is confusing and very long, it is not clearly understood the purpose of the work and the ways in which the objectives are achieved.
- The representation of numerical values are often wrong.
-Please pay attention to the decimals. In some cases are presented numbers with three decimals that are not compatible with the accuracy of the instruments.
- Please pay attention to the units. Especially in the figures, the use of capital letters should be checked.

SECTION 1 and 2
-I suggest revising the introduction (sections 1 and 2) and better explain the aim of the study. (The objectives of the research are mentioned at pages 15, line 454-456)
The sentence "This paper adds value in several ways; it contributes theoretically, methodologically, as well as empirically" (Line 126-127) is not sufficiently supported by the rest of the paper. What does it theoretically contribute to?
Section 2 is too long. I suggest more focused on the information strictly necessary to support the study.
Figure 3 - It is necessary? I believe that it does not add anything to the paper.

-Section 2.1.2. This part is too long, it might be better to collect all the Regulation and Standard information in a single summary table.

SECTION 3. Methodology
-Section Methodology should be completely revised. The section starts with the analysis of the climatic data but does not explain the procedure followed by the authors to reach their aims. It is not clear the use of the model, and the role of the simulation (as mentioned in Figure 11).
-I suggest inserting at the beginning of this section a summary of the activities carried out in order to guide the readers in the following pages.
-Figure 5 -Symbols represented in the legend do not appear in the graph. Please pay attention to the units (m/s and not M/S).
-Figure 6 Wind rose - The radar represents two information month and frequency. However, in a wind rose is expected to find the wind directions. I may have misunderstood myself, anyway please better explain the graph or properly modify the figure.
-Figure 8 - Please check the use of decimals and the letters used for the units.
-Figure 9 - The double representation of data is redundant. Please use or only the graph or only the table.
-Table 1 -I suggest to change the title "Description" with "Purpose"
-Figure 11 - The Letters A,B,C,D,E,F what do they indicate?
- Measurements conditions are not sufficiently described. During the indoor measurements, were the measurements carried out in free-run conditions? Or how many occupants were there? what behaviors did they follow during the monitoring period? (for example opening of the windows) Were there air conditioning systems inside the measured environment?
-Figure 13-16 The images of the instruments are not necessary. They might be useful if they capture a real measurement condition, on the contrary, it is sufficient to indicate the manufacturer and the model of the used instruments.

SECTION 4. Results and findings
-The evaluation of thermal comfort and the implications of the obtained results on the occupants' thermal perception are very interesting and it should be better commented. Furthermore, the parameters used to the calculus of the PMV should be added in the text (metabolic activity, basic clothing insulation, and eventually the correction of basic clothing insulation values).
-The text at line 504-511 is not results and it should be moved in the methodology part.
-The data represented in Figure from 17 to 22 could be merged. In order to allow easier comparison between the obtained results, I suggest replacing the figures with one or more tables.
-Tables 2,3 and Tables 4,5,6,7 should be merged into two tables.

7-Conclusions should be strongly revised.

Other minor corrections:
- The abstract is too long; in the instructions for authors the maximum length indicated is 200 words.
- Line 55-57 The sentence is not clear. Please better explain the requirements of 10% of the floor area (I think is referred to the area of the windows but it is not specified).
- Line 148 The number of the title is wrong.

Reviewer 3 Report

Overall, this is an interesting topic that potentially adds to the existing knowledge of building passive solutions in a hot-humid climate. However, some flaws affect the quality of the paper. In the following paragraphs, I discuss the strengths and weaknesses of this paper.

Abstract:

The abstract is informative and provides all the information. However, it is too long (407 words). It needs to be shortened and more to the point.

Introduction:

Overall, the introduction flow is smooth and brings up the research gap in a logical manner. However, it looks a bit wordy and needs to become shorter.

This is also the case for the literature background. This part also needs to be shortened substantially. More specifically, the main purpose of the research seems neglected and some introductory and unnecessary arguments with regard to the layout, openings, regulations, etc. are provided instead of providing more specific information. I recommend cutting down these parts and focus more on the main purpose of the research.

Please be consistent in units: use either Imperial or metric for all the measures.

Methodology:

In the first paragraph, there is a need for an introductory paragraph about the overall method used in this study in order to give some heads-up to the reader about what they should expect. This is very important since it provides a sense of direction for the reader.

Using only one case study cannot provide valid and reliable results. According to the categorization mentioned in this manuscript, there are several types of terraced houses each of which having unique characteristics. Therefore, in order for such a study to provide valid and reliable results, multiple case studies should be considered.

Results and finding,

I found this section informative (regardless of the number of case studies) yet wordy. It is expected to summarize the results in a way that directs the reader to a relevant discussion. Although the findings are discussed in each sub-section, they are not organized and well-presented as the outcome of a scientific study. It is recommended to provide a section specifically for the discussion.

Conclusion:

The findings and potential future studies have been discussed in this section. However, limitations of the study should be discussed in this section (for example, studying only one case is the main limitation that needs to be acknowledged).

Round 2

Reviewer 1 Report

Presenting % deviation in temperature reading in °C is misleading as °C is not an absolute unit. Therefore, if you change the unit to °F, the % deviation values would also change. Therefore, suggest removing % deviation for temperature data.

For example, it is not correct to say that 25°C is 25% greater than 20°C. Because, if you convert the temperature to absolute T, 25°C (298K) is only 2% greater than 20°C (293K).  

Reviewer 2 Report

The authors presented a version of the article very similar to the previous one. Although in the replies to the reviewer, they wrote that they had made all the required changes, in reality, they only modified the manuscript to a small extent. For example old Figure 8 (now Figure 7) has not been modified, the abstract is still too long, PMV input data have not been added, etc.
In this case, I would have preferred reasoned responses, rather than an apparent acceptance of the revision.
For this reason, I believe that the article is not suitable for publication in an international journal.

Reviewer 3 Report

Dear author(s):

The revision has significantly improved the quality of your manuscript. However, there are still some comments that need your consideration.

Abstract is still too long and needs to be cut down to 250 words maximum.

The limitation(s) of the study is not still discussed in the "Conclusion" section.